# Design and Synthesis of N-Substituted 3,4-Pyrroledicarboximides as Potential Anti-Inflammatory Agents

**DOI:** 10.3390/ijms22031410

**Published:** 2021-01-30

**Authors:** Aleksandra Redzicka, Żaneta Czyżnikowska, Benita Wiatrak, Katarzyna Gębczak, Andrzej Kochel

**Affiliations:** 1Department of Chemistry of Drugs, Wroclaw Medical University, Borowska 211, 50-556 Wrocław, Poland; 2Department of Inorganic Chemistry, Faculty of Pharmacy, Wroclaw Medical University, Borowska 211a, 50-556 Wroclaw, Poland; zaneta.czyznikowska@umed.wroc.pl; 3Department of Basic Medical Sciences, Wroclaw Medical University, Borowska 211, 50-556 Wrocław, Poland; benita.wiatrak@umed.wroc.pl (B.W.); katarzyna.gebczak@umed.wroc.pl (K.G.); 4Department of Pharmacology, Faculty of Medicine, Wroclaw Medical University, Mikulicza-Radeckiego 2, 50-345 Wrocław, Poland; 5Department of Chemistry, University of Wroclaw, F. Joliot-Curie 14, 54-234 Wrocław, Poland; andrzej.kochel@chem.uni.wroc.pl

**Keywords:** pyrrolo[3,4-*c*]pyrrole, cyclic imides, COX-1/COX-2 inhibition, Mannich bases, analgesic activity, inflammatory agents, docking study

## Abstract

In the present paper, we describe the biological activity of the newly designed and synthesized series N-substituted 3,4-pyrroledicarboximides **2a**–**2p**. The compounds **2a**–**2p** were obtained in good yields by one-pot, three-component condensation of pyrrolo[3,4-*c*]pyrrole scaffold (**1a**–**c**) with secondary amines and an excess of formaldehyde solution in C_2_H_5_OH. The structural properties of the compounds were characterized by ^1^H NMR, ^13^C NMR FT-IR, MS, and elemental analysis. Moreover, single crystal X-ray diffraction has been recorded for compound **2h**. The colorimetric inhibitor screening assay was used to obtain their potencies to inhibit COX-1 and COX-2 enzymes. According to the results, all of the tested compounds inhibited the activity of COX-1 and COX-2. Theoretical modeling was also applied to describe the binding properties of compounds towards COX-1 and COX-2 cyclooxygenase isoform. The data were supported by QSAR study.

## 1. Introduction

An important group of scaffolds for the synthesis of bioactive compounds such as anti-microbial [1,2,3,4,5,6,7,8,9,10,11,12,13,14,15], antitumor [16,17,18,19,20,21,22,23], antihyperlipidemic, hypoglycemic [24,25], and carbonic anhydrase inhibitors [26,27,28,29,30,31] are cyclic imides. The imide core is also present in compounds whose activity is related to CNS activity [32,33,34,35,36]. In particular, much attention has been paid to their ability to inhibit cyclooxygenases, anti-inflammatory, and analgetic activity [37,38,39,40,41,42,43,44,45,46,47]. To date, several drugs with cyclic imide structure have been approved, including lurasidone, buspirone, ethosuximide, mesuximide, and the IMiDs class includes thalidomide and its analogues lenalidomide, pomalidomide, apremilast (Figure 1).

The structure of many significant natural products, for example cladoniamide A, lamprolobine, migrastatine, julocrotine, is based on the cyclic imide motif [48]. One of the natural precursors used in the synthesis of imide compounds is phyllanthimide isolated from leaves of *Phyllanthus sellowianus* (*Euphorbiaceae*) [49]. The properties of the imide derivatives are strongly dependent on the nature of substituent in the imide ring. For example, the size and electrophilic character of substituents influence their steric properties [50]. The presence of specific groups with nitrogen and oxygen atoms is responsible for pharmacological consequences [51,52]. Additionally, imide based compounds are often neutral and hydrophobic, which determines their ability to penetrate biological membranes [9,50,53].

We have recently reported on the synthesis and COX-1/COX-2 inhibition of N-substituted cyclic imides (Figure 2) [39].

The obtained structures (Figure 2) inhibited less COX-2 and more COX-1 than the reference drug—meloxicam. Therefore, their selectivity ratio (COX-2/COX-1) was more than twice as high as meloxicam [39].

Based on the fragmental data in this paper, we undertook further modifications of the 4-arylpiperazinyl fragment (Figure 2) in order to increase COX-2 inhibitions. Structural modifications were achieved by varying the 4-substituent on the piperazine moiety and the synthesis of analogues with 4-arylpiperidine or morpholine pharmacophore for the structure–activity relationship (SAR) studies. In all the new compounds, we retained the one-carbon central alkyl chain as an optimal spacer.

The new set of compounds was investigated in terms of their activity as potential COX-1 and COX-2 inhibitors by colorimetric inhibitor screening assay [54]. Moreover, the binding mode of designed derivatives was obtained based on molecular docking results and previously reported data.

## 2. Results and Discussion

### 2.1. Chemistry

The new derivatives **2a**–**2p** were obtained as a result of a one-step synthesis, as shown in Scheme 1. The key intermediates, imides **1a**–**1c**, were prepared from ethyl α,β-diacetylsuccinate in a five-step synthesis according to known methods [39,55]. The preparation of the final compounds **2a**–**2p** involved condensation of the substrate **1a**-**1c** and the appropriate derivatives of 1-substituted piperazine **2a**–**2j** (arylpiperazines **2a**–**2e**, heteroarylpiperazine **2f**, cyclohexylpiperazines **2g**–**2j**, hydroxyethylpiperazine **2j**), piperazine **2k**, morpholine **2l** and 4-arylpiperidine **2m**–**2p** (Scheme 1).

The reactions were carried out under mild conditions and gave, in several cases, the desired products in high yield (70–84%). The piperazine, piperidine, and morpholine intermediates were commercially available with at least 97% purity.

The crude products were purified by crystallization from ethanol. The identity of the new compounds was established by their FT-IR, ^1^H NMR, ^13^C NMR, elemental analyses, and MS. The ^1^H NMR spectra of **2a**–**2p** revealed that the presence of the methylene group of the Mannich base showed at δ∼4.5 ppm (s, 2H). The obtained values of elemental analysis were in consonance with theoretical information. Additionally, the structure of **2h**, taken as an example, was unambiguously established by X-ray crystallography (Figure 3). 

The crystal data and structure refinement for **2h** are summarized in Table 1.

In general, the bond lengths and angles do not exhibit surprising features [56].

In the crystal lattice, a C-H…O type hydrogen bond is present which can additionally stabilize the crystal lattice (see Appendix A). The angle in the N21-C20-N2 molecule is equal to 117.1(2)° and therefore the molecule is generally V-shaped. A similar particle shape was observed for newly designed pyrrolo[3,4-*c*]pyrrole derivatives, where the angle in the N12-C11-N2 molecule was equal to 116.4(5)° [39]. In addition, the presence of atom Cl in the benzene ring causes it to align itself perpendicularly to the flat core of the pyrrolo[3,4-*c*]pyrrole.

### 2.2. Biological Tests 

#### 2.2.1. Cyclooxygenase Inhibition

The researched compounds inhibited both the COX-1 and the COX-2 enzymes (Table 2). Compounds **2b** and **2c** had similar COX selectivity ratios and showed stronger activity than meloxicam, which was used as a reference, in the same concentration. The compounds **2a**–**2c**, **2j**, and **2m** blocked the COX-1 isoform more strongly than the reference compound. In the case of the COX-2 enzyme, all tested structures displayed higher activity than the meloxicam.

#### 2.2.2. Viability of Cell Cultures

All tested compounds **2a**–**2p** showed no cytotoxic activity in the tested concentration range (Table 2). The viability of NHDF cells decreased with increasing concentration, but only for compound **2g**, the calculated IC_50_ value was lower than the highest concentration used. Based on the IC_50_ values, it can be assumed that the tested compounds may show a pro-proliferative effect on human dermal fibroblasts (NHDF cells).

### 2.3. Computational Studies

#### 2.3.1. The Analysis of Lipinski and Veber Rules

In the next step, using the SwissAdme website [57] physicochemical properties of the final compounds **2a**–**2p** were determined based on Lipinski and Veber rules (Table 3).

The Lipinski and Veber rules are used to evaluate drug-likeness or to determine if compounds possess chemical and physical properties to be an orally active drug in humans. Compounds that do not meet at least two of the criteria of the Lipinski rules may cause absorption or permeability problems. The criteria of Lipinski rules are molecular weight (MW) ≤ 500 Da, lipophilicity values (log P) ≤ 5, number of hydrogen bond donors (NHD) ≤ 5, number of hydrogen bond acceptors (NHA) ≤ 10. Veber rules are rotatable bonds (NBR) ≤ 10 and polar surface area (PSA) ≤ 140 Å^2^ [58,59].

The 14 tested compounds comply with Lipinski’s rule of five and contained less than five hydrogen bond donors, less than 10 hydrogen bond acceptors, molecular weights below 500 Da, and logP values < 5. Moreover, in line with the Veber rule, these molecules had the number of rotating bonds (NBR) less than 10 and polar surface area (PSA) values lower than 140 Å^2^.

#### 2.3.2. Structure–Activity Relationship (SAR) of N-Substituted 3,4-Pyrroledicarboximides **2a**–**2p**

Despite the limited number of compounds tested, on the bases of the results presented in Table 2, we formulated some SAR.

For SAR discussion, the compounds were divided into two series—derivatives of piperazine (**2a**–**2k** and **2l**) and piperidine **2m**–**2p**. We included compound **2l** in the piperazine group because we treated it as an analogue of a **2k** compound in which an oxygen atom replaced the nitrogen atom in the piperazine ring.

The SAR of N-substituted 3,4-pyrroledicarboximides (Table 2) shows the following:All compounds **2a**–**2p** inhibited COX-2 a stronger than meloxicam, which was used as a reference.Within the piperazine derivatives the best COX selectivity ratio IC_50_ (COX-2)/IC_50_ (COX-1) had compounds **2b** and **2c**. Compound **2b** does not meet the Lipinski and Veber rules. However, it should be emphasized that these rules define only general properties that describe potential orally administered drugs. Therefore, the predictive ability of these rules may be limited in some cases, and one should be careful during interpretation such type of results.Replacement of aryl substituent in a series of piperazine derivatives with heteroaryl or cycloalkyl substituent (**2g**–**2i**) leads to a significant reduction in COX-1 inhibition while maintaining COX-2 inhibition higher than that of the reference drug. The **2h** compound has the lowest COX-1 inhibition value.Introduction of hydrophilic hydroxyethyl substituent (**2j**) leads to a decrease of inhibitory activity in relation to both enzymes.Replacement of the 4-arylpiperazine moiety with 4-arylpiperidine leads to a loss of COX-2 inhibition.Introduction of group OH in a series of arylpiperidine derivatives significantly reduces the COX-1 inhibition (compare **2m** to **2n**–**2p**).

#### 2.3.3. Molecular Docking Studies

Molecular docking simulations were performed in order to predict the binding mode of all designed compounds to the main binding sites of both enzymes. The obtained data were presented in Figure 4 and Figure 5 in the manuscript and Appendix A (**S2a**–**S2p**). 

The results of our study show that most of the tested inhibitors (**2a**–**2c**, **2f**–**2i**, **2m**–**2n**, **2o**–**2p**) are more effective against COX-2, which is consistent with the results of enzymatic measurements. In the mentioned cases, the binding manner to the active center of COX-2 and the potency of binding are similar. Most of them can bind to the additional binding pocket of COX-2 including Leu352, Ser353, Tyr355, Phe518, and Val523 amino acid residues which arise due to the conformation of Tyr355 induced by the presence of Val523 instead of Ile523.

Compounds **2g**–**2i** containing a cyclohexylpiperazine substituent take the same orientation in the binding site of an enzyme (Figure 4 and Figure 5). All derivatives show significant inhibitory activity towards COX-2, among which **2h** demonstrates to be the most potent compound (inhibition constant 1.84 nM). Binding free energy of the most stable complex (**2h** - COX-2) was established as −11.9 kcal/mol (Table 4). All compounds can form two hydrogen bonds, between the oxygen of pyrrolo[3,4-*c*]pyrrole moiety and Tyr355 and Arg120 residues. As determined, cyclohexylpiperazine rings form hydrophobic interactions with Val116, Leu117, Ile345, Leu531, and Leu359 amino acids. Benzene and chlorobenzene rings of **2g** and **2h** are exposed towards hydrophobic and polar amino acid residues (Leu352, Trp387, Ala527, and Ser530), whereas the aliphatic substituent of **2i** is located near Val522, Leu352, Ser352, Met522, and Ala527 of COX-2. 

The presence of a hydroxy pyridine ring (**2n**–**2p**) strongly affects the way of binding. In this case, the strongest inhibitor of COX-2 is compound **2n** (free energy of binding −10.3 kcal/mol). Two hydrogen bonds are created between the hydroxyl pyridine ring of **2n** and Asp120 and Tyr355 amino acid residues and hydrophobic interactions with Val89, Val116, Arg120, Leu359, and Tyr355. According to a molecular docking study compound, **2p** is a more potent inhibitor of COX-2 in terms of free energy of binding (−10 kcal/mol). There is also the possibility of H-bonding interactions which involved oxygen atom of pyrrolo[3,4-*c*]pyrrole moiety and Tyr355 of COX-2. In the vicinity of the aliphatic chain of **2p** hydrophobic and polar amino acids are present (Leu352, Leu384, Tyr385, Trp387, Met522, Val532, and Gly526). Compound **2o** exhibits a unique binding configuration in the binding site of COX-2. As presented in Figure 6 it can form two hydrogen bonds with Val116 and Ser530, and the hydroxyl pyridine group forms hydrophobic interactions Leu93, Val116, Arg120, Leu359, and Ala527. Additionally, an aliphatic chain of **2o** forms hydrophobic interactions with Leu352, Leu384, Trp387, Phe518, Met522, Gly526. 

Compounds **2a**–**2c**, **2f**, **2m** are selective inhibitors of COX-2 (binding energy ranges from −10.4 to −11.3 kcal/mol). In the case of **2a**, one hydrogen bond with Ser530 is formed. Compound binding to the active site of COX-2 is exposed towards hydrophobic amino acids residues (Leu93, Val116, Val349, Leu352, Met522, Val523, Gly526, Ala527), which are mainly involved in van der Waals type interactions. The obtained results show that compound **2f** is almost located in the same binding place as meloxicam and is a more effective inhibitor of COX-2 (inhibition constant 5.57 nM). In this case, it occupies an additional binding cavity of COX-2 and is located in close proximity of Leu352, Tyr355, Phe518, and Val523.

The data indicate that compounds **2d**, **2j**, **2l**, **2e**, and **2k** are not selective inhibitors of cyclooxygenases. Although the binding manner to both enzymes differs significantly, the free energy of binding ranges from −8.2 to −9.7 kcal/mol (Table 4 and Table 5).

Compound **2d** forms hydrogen bonding three hydrogen bonds with Lys360, Asp352, Leu531 and Ser530, Leu531, Met535 of COX-1 and COX-2, respectively. The aliphatic substituent of **2d** was observed to be encompassed by a number of polar and hydrophobic amino acids of COX-1, namely, Leu352, Trp387, Phe518, Met 522, Ile523, Gly526, and Ser530. In the case of COX-2 in close proximity of chain are additionally located Leu384, Tyr385, and Val523 instead of Ile523. 

The replacement of benzene rings with aliphatic chains decreased the affinity of compound **2j** to cyclooxygenases. In both cases docking study showed hydrophobic interactions with Ile345, Val349, Leu352, Trp387, Met522, Gly526, Ala527, Ser530, Leu534 (see Appendix A). The orientation of **2l** in the binding sites of COX-1 and COX-2 is practically the same. In both cases one hydrogen bonds between compound and oxygen atom od -OH group of Ser530. Benzene moiety occupied the binding pocket of enzymes formed by Leu384, Trp387, Phe518, Met522, and Gly526. Moreover, the morpholine ring occupied the hydrophobic pocket formed by Met113, Val349, Leu531, Leu534, Leu535 in the case of COX-1 and Met113, Leu117, Ile345, Leu531 in the case of COX-2. 

#### 2.3.4. QSAR Analysis of Biological Properties

The biological activity of all designed compounds was estimated using 3D/4D QSAR model with restricted docking protocol and was expressed as the probability to be active. Analgetic activity, anti-inflammatory activity in vivo oedema paw carrageenin, anti-inflammatory activity combined action, antioxidant activity, LOX inhibitory activity were evaluated. The data are presented in Appendix A. Compounds **2o**, **2p**, and **2e** show anti-oxidant activity (over 60%). The very high probability of LOX inhibition exhibit compounds **2d**, **2h**, **2f**, **2c**, **2b**, **2m**, **2j**, **2p**, and **2e**. Most of the synthesized derivatives have anti-inflammatory activity, additionally compounds **2b**–**2f**, **2n**, and **2m** can be used as analgetic agents. 

## 3. Materials and Methods

### 3.1. Chemistry

#### 3.1.1. Instrumentation and Chemicals

All chemicals, reagents, and solvents used in the current study were purchased from commercial suppliers (Chemat, Gdańsk, Poland; Archem, Łany, Poland; Alchem Wrocław, Poland) and used without further purification. Dry solvents were obtained according to the standard procedure. Progress of the reaction was monitored by the thin-layer chromatography (TLC) technique on silica-gel-60-F254-coated TLC plates (Fluka Chemie GmbH, Switzerland) and visualized by UV light at 254 nm. The melting points of received products were determined by an open capillary method on Electrothermal Mel-Temp 1101D apparatus (Cole-Parmer, Vernon Hills, IL, USA) and were uncorrected. The ^1^H NMR (300 MHz) and ^13^C NMR (75 MHz) spectra were recorded on a Bruker 300 MHz NMR spectrometer (Bruker Analytische Messtechnik GmbH, Rheinstetten, Germany) in CDCl_3_ using tetramethylsilane (TMS) as an internal reference. Spectral data includes chemical shifts in ppm, multiplicities, constant couplings in Hz, number of protons, protons’ positions. Multiplicities are abbreviated as follow: s (singlet), d (doublet), t (triplet), and m (multiplet). Spectra were recorded and read using TopSpin 3.6.2 Bruker Daltonik, GmbH, Bremen, Germany). The infrared (IR) spectra were determined on a Nicolet iS50 FT-IR Spectrometer (Thermo Fisher Scientific, Waltham, MA, USA). Samples were applied as solids, and frequencies are reported in cm^−1^. Spectra were read using OMNIC Spectra 2.0 Thermo Fisher Scientific, Waltham, MA, USA). Mass spectra were recorded using a Bruker Daltonics Compact ESI-mass spectrometer (Bruker Daltonik, GmbH, Bremen, Germany). The instrument was operated in positive ion mode. Analyzed compounds were dissolved in a mixture of chloroform and methanol. Elemental analyses for carbon, nitrogen, and hydrogen were carried out on a Carlo Erba NA-1500 analyzer (Thermo Fisher Scientific, Waltham, MA, USA), and obtained results were within ± 0.4% of the theoretical values calculated for corresponding formulas. 

#### 3.1.2. General Procedure for Preparation of N-Substituted 3,4-Pyrroledicarboximides **2a**–**2p**

A solution of 2 mmol of pyrroledicarboximide **1a**–**1c** [39,55], 0.4 mL of 37% formaldehyde (*w*/*v*), and 2 mmol of appropriate amine in ethanol (25 mL) was refluxed for 3–5 h. The course of the reaction was controlled by TLC. The reaction was cooled, and the precipitate was filtered off. The crude product was purified by crystallization in ethanol.

4,6-dimethyl-5-phenyl-2{[4-(*p*-bromophenyl)-1-piperazinyl]methyl}pyrrolo[3,4-*c*]pyrrole-1,3(*2H*,*5H*)-dione *(***2a**).
**2a**: from **1a** and 1-(*p*-bromophenyl)piperazine. Yield 72%, m.p. 213–215 °C.**^1^H NMR** (300 MHz, CDCl_3_) δ: 2.18, 2.20 (2x s, 6H, 4,6–CH_3_), 2.75–2.85 (m, 4H, 2xCH_2_-piperazine), 3.15–3.25 (m, 4H, 2xCH_2_-piperazine), 4.58 (s, 2H, CH_2_), 6.73 (d, 2H, ArH, *J* = 9 Hz), 7.20–7.23 (m, 2H, ArH), 7.30–7.33 (m, 2H, ArH), 7.52–7.54 (m, 3H, ArH)**^13^C NMR** (75 MHz, CDCl_3_) δ: 166.05, 135.88, 131.84, 129.94, 129.64, 127.80, 117.75, 116.31, 58.31, 50.28, 48.99, 11.89**FT-IR** (selected lines, ϒ_max_, cm^−1^): 1688 (C=O), 1747 (C=O)**ESI-MS** (*m*/*z*): calcd. for C_25_H_25_BrN_4_O_2_ [M+H]^+^: 494.4034; found: 494.3202

4,6-dimethyl-5-(*o*-chlorophenyl)-2-{[4-(*p*-bromophenyl)-1-piperazinyl]methyl}pyrrolo[3,4-*c*]pyrrole-1,3(*2H*,*5H*)-dione (**2b**).
**2b**: from **1c** and 1-(*p*-bromophenyl)piperazine. Yield 78%, m.p. 230–232 °C.**^1^H NMR** (300 MHz, CDCl_3_) δ: 2.19, 2.22 (2x s, 6H, 4,6–CH_3_), 2.80–2.90 (m, 4H, 2xCH_2_-piperazine), 3.15–3.30 (m, 4H,2xCH_2_-piperazine), 4.58 (s, 2H, CH_2_), 6.73–6.76 (m, 2H, ArH), 7.11–7.15 (m, 1H, ArH), 7.25–7.32 (m, 3H, ArH), 7.50–7.53 (m, 2H ArH)**^13^C NMR** (75 MHz, CDCl_3_) δ: 165.82, 150.23, 137.00, 135.66, 131.84, 130.95, 130.04, 128.17, 126.17, 117.74, 116.62, 58.36, 50.28, 48.97, 11.87**FT-IR** (selected lines, ϒ_max_, cm^−1^): 1694 (C=O), 1752 (C=O)**ESI-MS** (*m*/*z*): calcd. for C_25_H_24_BrClN_4_O_2_ [M+H]^+^: 528.8485; found: 529.0797

4,6-dimethyl-5-phenyl-2{[4-(*o*-cyanophenyl)-1-piperazinyl]methyl}pyrrolo[3,4-*c*]pyrrole-1,3(*2H*,*5H*)-dione (**2c**).
**2c**: from **1a** and 1-(*o*-cyanophenyl)piperazine. Yield 70%, m.p. 203–205 °C. **^1^H NMR** (300 MHz, CDCl_3_) δ: 2.19, 2.25 (2x s, 6H, 4,6–CH_3_), 2.85–2.95 (m, 4H, 2xCH_2_-piperazine), 3.15–3.30 (m, 4H, 2xCH_2_-piperazine), 4.59 (s, 2H, CH_2_), 6.97–7.10 (m, 2H, ArH), 7.24–7.28 (m, 2H, ArH), 7.44–7.60 (m, 5H, ArH)**^13^C NMR** (75 MHz, CDCl_3_) δ: 166.03, 155.90, 136.00, 134.73, 133.76, 130.07, 129.89, 129.54, 127.87, 121.79, 118.87, 118.38, 116.42, 106.17, 58.26, 51.78, 50.55, 11.90**FT-IR** (selected lines, ϒ_max_, cm^−1^): 1694 (C=O), 1752 (C=O)**ESI-MS** (*m*/*z*): calcd. for C_26_H_25_N_5_O_2_ [M+H]^+^: 440.5168; found: 440.2015

4,6-dimethyl-5-phenyl-2{[4-(*m*-trifluoromethylphenyl)-1-piperazinyl]methyl}pyrrolo[3,4-*c*]pyrrole-1,3(*2H*,*5H*)-dione (**2d**).
**2d**: from **1a** and 1-(*m*-trifluoromethylphenyl)piperazine. Yield 82%, m.p. 232–235 °C. **^1^H NMR** (300 MHz, CDCl_3_) δ: 2.19 (s, 6H, 4,6–CH_3_), 2.83–2.86 (m, 4H, 2xCH_2_-piperazine), 3.23–3.26 (m, 4H, 2xCH_2_-piperazine), 4.58 (s, 2H, CH_2_), 7.02–7.08 (m, 3H, ArH), 7.21–7.24 (m, 2H ArH), 7.28–7.35 (m, 1H, ArH), 7.54–7.57 (m, 3H, ArH) **^13^C NMR** (75 MHz, CDCl_3_) δ: 166.07, 151.42, 135.92, 131.57, 131.16, 130.08, 129.92, 129.60, 129.49, 127.80, 126.12, 118.75, 116.37, 115.64, 115.59, 112.18, 112.13, 58.41, 50.36, 48.77, 11.86**FT-IR** (selected lines, ϒ_max_, cm^−1^): 1686 (C=O), 1745 (C=O)**ESI-MS** (*m*/*z*): calcd. for C_26_H_25_F_3_N_4_O_2_ [M+H]^+^ 483.5053; found: 483.1990

4,6-dimethyl-5-*n*-butyl-2-{[4-(*p*-methylphenyl)-1-piperazinyl]methyl}pyrrolo[3,4-*c*]pyrrole-1,3-(*2H*,*5H*)-dione (**2e**).
**2e**: from **1b** and 1-(*p*-methylphenyl)piperazine. Yield 80%, m.p. 115–117 °C.**^1^H NMR** (300 MHz, CDCl_3_) δ: 0.99 (t, 3H, CH_3_, *J* = 7.2 Hz), 1.37–1.44 (m, 2H, CH_2_), 1.62–1.68 (m, 2H, CH_2_), 2.26 (s, 3H, CH_3_), 2.41 (s, 6H, 4,6–CH_3_), 2.70–2.85 (m, 4H, 2xCH_2_-piperazine), 3.10–3.20 (m, 4H, 2xCH_2_-piperazine), 3.77 (t, 2H, CH_2_, *J* = 7.8 Hz), 4.54 (s, 2H, CH_2_), 6.80 (d, 2H, ArH, *J* = 9 Hz), 7.04 (d, 2H, ArH, *J* = 9 Hz)**^13^C NMR** (75 MHz, CDCl_3_) δ: 166.11, 129.58, 128.72, 116.60, 116.13, 58.23, 50.40, 49.82, 43.88, 32.45, 20.40, 20.02, 13.70, 11.35 **FT-IR** (selected lines, ϒ_max_, cm^−1^): 1694 (C=O), 1730 (C=O)**ESI-MS** (*m*/*z*): calcd. for C_24_H_32_N_4_O_2_ [M+H]^+^: 409.5443; found: 409.2565

4,6-dimethyl-5-*n*-butyl-2-{[4-(2-pyrimidinyl)piperazin-1-yl]methyl}pyrrolo[3,4-*c*]pyrrole-1,3(*2H*,*5H*)-dione (**2f**).
**2f**: from **1b** and 1-(2-pyrimidinyl)piperazine. Yield 71%, m.p. 203–205 °C.**^1^H NMR** (300 MHz, CDCl_3_) δ: 0.97 (t, 3H, CH_3_, *J* = 7.5Hz), 1.24–1.44 (m, 2H, CH_2_), 1.57–1.67 (m, 2H, CH_2_), 2.38 (s, 6H, 4,6–CH_3_), 2.65–2.71 (m, 4H, 2xCH_2_-piperazine), 3.75 (t, 2H, CH_2_, *J* = 7.5Hz), 3.80–3.82 (m, 4H, 2xCH_2_-piperazine), 4.52 (s, 2H, CH_2_), 6.43 (t, 1H, 5-H pyrimidine, *J* = 4.5 Hz), 8.25 (d, 2H, 4,6-H pyrimidine, *J* = 4.8 Hz)**^13^C NMR** (75 MHz, CDCl_3_) δ: 165.93, 161.56, 157.63, 128.72, 116.06, 109.65, 58.42, 50.46, 43.86, 43.50, 32.43, 19.99, 13.68, 11.34**FT-IR** (selected lines, ϒ_max_, cm^−1^): 1689 (C=O), 1738 (C=O)**ESI-MS** (*m*/*z*): calcd. for C_21_H_28_N_6_O_2_ [M+H]^+^ 397.4938; found: 397.2313

4,6-dimethyl-5-phenyl-2-[(4-cyclohexyl-1-piperazinyl)methyl]pyrrolo[3,4*-c*]pyrrole-1,3(*2H*,*5H*)-dione (**2g**).
**2g**: from **1a** and 1-cyclohexylpiperazine. Yield 75%, m.p. 140–142 °C.**^1^H NMR** (300 MHz, CDCl_3_) δ: 1.21–1.26 (m, 6H, 3xCH_2_), 1.62–1.88 (m, 4H, 2xCH_2_), 2.16 (s, 6H, 4,6–CH_3_), 2.50–2.65 (m. 4H, 2xCH_2_-piperazine), 2.70–2.74 (m, 4H, 2xCH_2_-piperazine), 4.52 (s, 2H, CH_2_), 7.19–7.26 (m, 2H, ArH), 7.54–7.56 (m, 3H, ArH)**^13^C NMR** (75 MHz, CDCl_3_) δ: 165.95, 137.09, 135.59, 130.91, 129.62, 128.19, 126.24, 116.83, 63.50, 58.27, 50.93, 48.89, 28.86, 26.23, 25.84, 11.78**FT-IR** (selected lines, ϒ_max_, cm^−1^): 1691 (C=O), 1746 (C=O)**ESI-MS** (*m*/*z*): calcd. for C_25_H_32_N_4_O_2_ [M+H]^+^ 421.5549; found: 421.2578 

4,6-dimethyl-5-(*o-*chlorophenyl)-2-[(4-cyclohexyl-1-piperazinyl)methyl]pyrrolo[3,4*-c*]pyrrole-1,3(*2H*,*5H*)-dione (**2h**).
**2h**: from **1c** and 1-cyclohexylpiperazine. Yield 72%, m.p. 199–201 °C.**^1^H NMR** (300 MHz, CDCl_3_) δ: 0.99–1.24 (m, 4H, CH_2_), 1.59–1.63 (m, 2H, CH_2_), 1.76–1.87 (m, 4H, CH_2_), 2.18 (s, 6H, 4,6–CH_3_), 2.50–2.60 (m. 4H, 2xCH_2_-piperazine), 2.65–2.71 (m, 4H, 2xCH_2_-piperazine), 4.53 (s, 2H, CH_2_), 7.14–7.15 (m, 1H, ArH), 7.25–7.27 (m, 1H, ArH), 7.51–7.55 (m, 2H ArH)**^13^C NMR** (75 MHz, CDCl_3_) δ: 165.96, 137.11, 135.59, 130.90, 129.95, 129.58, 128.19, 126.24, 116.85, 63.43, 58.33, 50.97, 48.92, 28.96, 26.31, 25.88, 11.77**FT-IR** (selected lines, ϒ_max_, cm^−1^): 1695 (C=O), 1756 (C=O)**ESI-MS** (*m*/*z*): calcd. for C_25_H_31_ClN_4_O_2_ [M+H]^+^: 455.9900; found: 455.2193

4,6-dimethyl-5-*n-*butyl-2-[(4-cyclohexyl-1-piperazinyl)methyl]pyrrolo[3,4-*c*]pyrrole-1,3*(2H*,*5H*)-dione (**2i**).
**2i**: from **1b** and 1-cyclohexylpiperazine. Yield 70%, m.p. 103–105 °C.**^1^H NMR** (300 MHz, CDCl_3_): δ: 0.98 (t, 3H, CH_3_, *J* = 7.2 Hz), 1.20–1.25 (m, 4H, 2xCH_2_), 1.35–1.42 (m, 2H, CH_2_), 1.60–1.63 (m, 4H, 2xCH_2_), 1.76–1.95 (m, 4H, 2xCH_2_), 2.37 (s, 6H, 4,6–CH_3_), 2.40–2.72 (m. 8H, 4xCH_2_-piperazine), 3.75 (t, 2H, CH_2_, *J* = 7.2 Hz), 4.47 (s, 2H, CH_2_)**^13^C NMR** (75 MHz, CDCl_3_) δ: 166.17, 129.12, 116.18, 43.85, 32.43, 28.85, 25.76, 20.02, 13.69, 11.41, 11.32**FT-IR** (selected lines, ϒ_max_, cm^−1^): 1685 (C=O), 1743 (C=O)**ESI-MS** (*m*/*z*): calcd. for C_23_H_36_N_4_O_2_ [M+H]^+^: 401.5653; found: 401.2901 

4,6-dimethyl-5-*n-*butyl-2-{[4-hydroxyethyl-1-piperazinyl]methyl}pyrrolo[3,4*-c*]pyrrole-1,3(*2H*,*5H*)-dione (**2j**).
**2j**: from **1b** and 1-(2-hydroxyethyl)piperazine. Yield 80%, m.p. 129–131 °C.**^1^H NMR** (300 MHz, CDCl_3_) δ: 0.99 (t, 3H, CH_3_, *J* = 7.2 Hz), 1.34–1.41 (m, 2H, CH_2_), 1.60–1.65 (m, 2H, CH_2_), 2.41 (s, 6H, 4,6–CH_3_), 2.60–2.65 (m, 6H, 2xCH_2_-piperazine+ CH_2_), 2.70–2.77 (m, 4H, 2xCH_2_-piperazine), 3.65 (t, 2H, CH_2_, *J* = 5.4Hz), 3.74 (t, 2H, CH_2_, *J* = 7.8 Hz), 4.48 (s, 2H, CH_2_)**^13^C NMR** (75 MHz, CDCl_3_) δ: 165.99, 128.89, 116.05, 59.90, 57.91, 56.96, 53.07, 49.27, 43.91, 32.46, 20.05, 13.69, 11.38**FT-IR** (selected lines, ϒ_max_, cm^−1^): 1685 (C=O), 1737 (C=O), 3199 (OH)**ESI-MS** (*m*/*z*): calcd. for C_19_H_33_N_5_O_3_ [M+H]^+^ 363.4743; found: 363.2362

4,6-dimethyl-5-phenyl-2-[(piperazinyl)methyl]pyrrolo[3,4*-c*]pyrrole-1,3(*2H*,*5H*)-dione (**2k**).
**2k**: from **1a** and piperazine. Yield 84%, m.p. 288–290 °C.**^1^H NMR** (300 MHz, CDCl_3_) δ: 2.19 (s, 6H, 4,6–CH_3_), 2.60–2.75 (m. 4H, 2xCH_2_-piperazine), 3.60–3.75 (m, 4H, 2xCH_2_-piperazine), 4.50 (s, 2H, CH_2_), 7.21–7.26 (m, 2H, ArH), 7.54–7.57 (m, 3H, ArH)**^13^C NMR** (75 MHz, CDCl_3_) δ: 166.02, 135.98, 129.89, 129.55, 127.85, 116.41, 58.27, 50.46, 11.84**FT-IR** (selected lines, ϒ_max_, cm^−1^): 1686 (C=O), 1747 (C=O)**ESI-MS** (*m*/*z*): calcd. for C_19_H_22_N_4_O_2_ [M+H]^+^ 339.4113; found: 339.4114

4,6-dimethyl-5-phenyl-2-[(morpholinyl)methyl]pyrrolo[3,4-*c*]pyrrole-1,3(*2H*,*5H*)-dione (**2l**).
**2l**: from **1a** and morpholine. Yield 73%, m.p. 133–135 °C.**^1^H NMR** (300 MHz, CDCl_3_) δ: 2.18 (s, 6H, 4,6–CH_3_), 2.60–2.75 (m, 4H, 2xCH_2_-piperazine), 3.60–3.75 (m, 4H, 2xCH_2_-piperazine), 4.50 (s, 2H, CH_2_), 7.21–7.26 (m, 2H, ArH), 7.54–7.57 (m, 3H, ArH)**^13^C NMR** (75 MHz, CDCl_3_) δ: 165.99, 136.00, 129.93, 129.61, 127.81, 116.35, 67.01, 58.71, 50.89, 11.88 **FT-IR** (selected lines, ϒ_max_, cm^−1^): 1687 (C=O), 1746 (C=O)**ESI-MS** (*m*/*z*): calcd. for C_19_H_21_N_3_O_3_ [M+H]^+^: 340.3961; found: 340.1622

4,6-dimethyl-5-*n-*butyl-2-[(4-phenyl-1-piperidinyl)methyl]pyrrolo[3,4*-c*]pyrrole-1,3(*2H*,*5H*)-dione (**2m**).
**2m**: from **1b** and 4-phenyl-1-piperidine. Yield 80%, m.p. 232–235 °C.**^1^H NMR** (300 MHz, CDCl_3_) δ: 0.99 (t, 3H, CH_3_, *J* = 7.5 Hz), 1.35–1.40 (m, 2H, CH_2_), 1.45–1.58 (m, 2H, CH_2_), 1.60–1.73 (m, 4H, CH_2_), 1.80–1.90 (m, 2H, CH_2_), 2.42 (s, 6H, 4,6–CH_3_), 3.10–3.21 (m, 4H, 2XCH_2_), 3.78 (t, 2H, CH_2_, *J* = 7.8 Hz), 4.51 (s, 2H, CH_2_), 7.10–7.39 (m, 5H, ArH)**^13^C NMR** (75 MHz, CDCl_3_) δ: 166.17, 128.53, 116.24, 48.81, 43.83, 32.43, 25.75, 20.00, 13.68, 11.30 **FT-IR** (selected lines, ϒ_max_, cm^−1^): 1686 (C=O), 1747 (C=O)**ESI-MS** (*m*/*z*): calcd. for C_24_H_40_N_4_O_2_ [M+H]^+^;394.8590 found: 394.2452

4,6-dimethyl-5-*n-*butyl-2-{[4-(*p-*bromophenyl)-4-hydroxypiperidin-1-yl]methyl}pyrrolo[3,4*-c*]pyrrole-1,3*(2H*,*5H*)-dione (**2n**).
**2n**: from **1b** and 4-(*p*-bromopheny)l-4-hydroxypiperidine. Yield 74%, m.p. 169–171 °C.**^1^H NMR** (300 MHz, CDCl_3_) δ: 1.00 (t, 3H, CH_3_, *J* = 7.2 Hz), 1.38–1.48 (m, 2H, CH_2_), 1.62–1.75 (m, 4H, CH_2_), 2.00–2.20 (m, 2H, CH_2_), 2.42 (s, 3H, 4–CH_3_), 2.43 (s, 3H, 6–CH_3_), 2.65–2.75 (m. 2H, CH_2_), 2.85–3.10 (m, 2H, CH_2_), 3.79 (t, 2H, CH_2_, *J* = 7.8 Hz), 4.54 (s, 2H, CH_2_), 7.20–7.32 (m, 2H, ArH), 7.42–7.45 (m, 2H, ArH)**^13^C NMR** (75 MHz, CDCl_3_) δ: 165.92, 146.57, 133.25, 128.38, 126.10, 46.88, 43.91, 32.48, 20.04, 13.69, 11.36**FT-IR** (selected lines, ϒ_max_, cm^−1^): 1680 (C=O), 1733 (C=O), 3524 (OH)**ESI-MS** (*m*/*z*): calcd. for C_24_H_30_BrN_3_O_3_ [M+H]^+^;489.4251 found: 489.1990

4,6-dimethyl-5-*n-*butyl-2-{[4-(*p-*chlorophenyl)-4-hydroxypiperidin-1-yl]methyl}pyrrolo[3,4*-c*]pyrrole-1,3*(2H,5H*)-dione (**2o**).
**2o**: from **1b** and 4-(*p*-chlorophenyl)-4-hydroxypiperidine. Yield 78%, m.p. 170–172 °C.**^1^H NMR** (300 MHz, CDCl_3_) δ: 0.98 (t, 3H, CH_3_, *J* = 7.2 Hz), 1.30–1.45 (m, 2H, CH_2_), 1.62–1.71 (m, 4H, 2xCH_2_), 2.00–2.20 (m, 2H, CH_2_), 2.38 (s, 6H, 4,6–CH_3_), 2.60–2.75 (m, 2H, CH_2_), 2.85–2.95 (m, 2H, CH_2_), 3.76 (t, 2H, CH_2_, *J* = 7.8 Hz), 4.49 (s, 2H, CH_2_), 7.33–7.36 (m, 2H, ArH), 7.42–7.46 (m, 2H, ArH)**^13^C NMR** (75 MHz, CDCl_3_) δ: 166.08, 146.87, 132.75, 128.75, 128.36, 126.11, 116.14, 98.74, 70.67, 58.51, 46.87, 43.90, 38.30, 32.47, 20.04, 13.70, 11.37 **FT-IR** (selected lines, ϒ_max_, cm^−1^): 1680 (C=O), 1733 (C=O), 3517 (OH)**ESI-MS** (*m*/*z*): calcd. for C_24_H_30_ClN_3_O_3_ [M+H]^+^: 444.9741; found: 444.2015

4,6-dimethyl-5-*n-*butyl-2-{[4-benzyl-4-hydroxypiperidin-1-yl]methyl}pyrrolo[3,4*-c*]pyrrole-1,3(*2H*,*5H*)-dione (**2p**).
**2p**: from **1b** and 4-benzyl-4-hydroxypiperidine, Yield 83%, m.p. 141–143 °C.**^1^H NMR** (300 MHz, CDCl_3_) δ: 0.98 (t, 3H, CH_3_, *J* = 7.2 Hz), 1.35–1.40 (m, 2H, CH_2_), 1.45–1.58 (m, 2H CH_2_), 1.60–1.73 (m, 4H, 2xCH_2_), 2.38 (s, 6H, 4,6–CH_3_), 2.50–2.60 (m, 2H, 2XCH_2_), 2.71 (s, 2H, CH_2_), 2.71–2.78 (m, 2H, CH_2_), 3.75 (t, 2H, CH_2_, *J* = 7.8 Hz), 4.45 (s, 2H, CH_2_), 7.15–7.18 (m, 2H, ArH), 7.22–7.29 (m, 3H, ArH)**^13^C NMR** (75 MHz, CDCl_3_) δ: 165.92, 136.37, 130.54, 128.75, 128.27, 126.63, 116.12, 68.72, 58.36, 48.88, 46.82, 43.88, 36.49, 32.46, 20.03, 13.69, 11.34**FT-IR** (selected lines, ϒ_max_, cm^−1^): 1682 (C=O), 1742 (C=O), 3504 (OH)**ESI-MS** (*m*/*z*): calcd. for C_25_H_33_N_3_O_3_ [M+H]^+^ 424.5556; found: 424.2565.

### 3.2. Crystallography

#### 3.2.1. X-ray Structure Determinations of **2h**

##### X-ray Experimental Details

Crystals of **2h** suitable for single-crystal X-ray diffraction analysis were obtained by dissolution in ethanol/2-propanol followed by slow evaporation of the solvent at room temperature. Crystallographic measurements for 1 was collected with Κ-geometry diffractometers: Xcalibur Gemini with graphite monochromatized Cu-Kα radiation (λ = 1.5418 Å) at 100(2) K, using an Oxford Cryosystems cooler. Data collection, cell refinement, data reduction, and analysis were carried out with CrysAlisPro [60]. Analytical absorption correction was applied to data with the use of CrysAlisPro. The crystal structures were solved using SHELXS [61] and refined on F2 by a full-matrix least squares technique with SHELXL-2015 [62] with anisotropic thermal parameters for all the ordered non-H atoms. In the final refinement cycles, H atoms were repositioned in their calculated positions and treated as riding atoms, with C-H = 0.95–0.98 Å, and with Uiso(H) = 1.2 Ueq(C) for CH, CH_2_ and 1.5 Ueq(C) for CH_3_. All figures were made using DIAMOND program [63]. CCDC 2036201 contain the supplementary crystallographic data for this paper. These data can be obtained free of charge via www.ccdc.cam.ac.uk/data_request/cif, by e-mailing data_request@ccdc.cam.ac.uk, or by contacting the Cambridge Crystallographic Data Centre, 12 Union Road, Cambridge CB2 1EZ, UK; fax: + 44(0)1223-336033.

### 3.3. Pharmacology

#### 3.3.1. Determination of COX-1/COX-2-Inhibition

##### Material and Methods

Cell Line

The NHDF cell line (from LONZA) was used to determine the cytotoxicity effect of newly synthesized compounds. The cells were incubated in DMEM without phenol red supplemented with 10% fetal bovine serum (FBS), 2 mM L glutamine, 1.25 µg/mL amphotericin B, and 100 µg/mL gentamicin. The medium prepared in this way was stored at 4–8 °C. The NHDF cells were cultivated at 37 °C in 5% CO_2_ and 95% humidity. The cells were passaged, or the medium was changed twice a week.

#### 3.3.2. Tested Compounds

The 15 newly synthesized compounds were dissolved in DMSO to the final concentration of 10 mM (stock solutions) and stored at −20 °C for up to 6 months. The working solution was dissolved in a culture medium to a final concentration of 10, 50, and 100 µM (the DMSO concentration did not reach 1%).

#### 3.3.3. SRB Assay

The SRB assay was performed to evaluate the cytotoxicity effect of newly synthesized compounds on NHDF cells. The cells were seeded at a density of 10,000 cells/well on the 96-well plates and left at 37 °C in the 5% CO_2_ incubator for 24 h to allow cells to adhere to the well surface. After removing the supernatant, tested compounds were applied for 48 h. The 10% *w*/*v* cold TCA solution was added to the culture plate for 1 h at 4–8 °C to fix the cells. The plates were washed four times with running water and air-dried at RT. The 0.4% SRB dye solution in 1% acetic acid was applied for 30 min at RT. The plates were then rinsed with 1% (*v*/*v*) acetic acid five times and air-dried again at RT. The 10 mM Trizma base was supplied to dissolve protein-bound dye for 30 min, and the absorbance was measured at 565 nm using Varioskan LUX microplate reader (Thermo Scientific).

#### 3.3.4. Cyclooxygenase Inhibition Assay

The COX peroxidase activity of all tested compounds at a concentration of 100 µM was assessed using a ready-to-use kit (Cayman, cat. no. 701050). The peroxidase absorbance was measured after 2 min incubation at RT using Varioskan LUX (λ = 590 nm). Based on the obtained results, the IC_50_ values (concentrations at which 50% inhibition of enzyme activity was observed) were calculated separately for COX-1 and COX-2 enzymes. There were also calculated the ratios of IC_50_ values (IC_50_(COX-2)/IC_50_(COX-1)), which allowed assessing the selectivity of cyclooxygenase inhibition [64].

#### 3.3.5. Statistical Analysis

All results were presented as IC_50_ values ± SD. All biological assays were performed in three independent replications. Significance was assessed with parametric tests—ANOVA and Tukey’s post-hoc. The significance level was assumed as *p* < 0.05.

### 3.4. Molecular Modeling

The structures of designed compounds were optimized at the CAM-B3LYP/6-31++G** level of theory with the polarizable continuum model (PCM) including solvent effects [65,66,67] using the Gaussian 09 program [68]. Molecular docking was performed using AutoDock4.2 package, and a standard protocol was followed to predict the binding mode and the free energy of binding [69]. The following equation expresses the free energy of binding, which characterizes the affinity of protein-ligand complexes:ΔG binding = [ΔG intermolecular + ΔG internal + ΔG tors] − ΔG unbound

The crystallographic structures of COX-1 (PDB ID: 4O1Z) and COX-2 (PDB ID: 4M11) co-crystallized with meloxicam originated from Protein Data Bank [70]. The polar hydrogen atoms and solvent parameters were added to the chain A of cyclooxygenases and Gasteiger charges for each of the atoms were declared. The binding place was defined using the grid of 80 × 80 × 80 points with 0.375 Å spacing. The grid center was established in the active site according to the crystalized inhibitor location. The Lamarckian genetic algorithm (LGA) implemented with local search and 200 runs for each complex was used during the docking procedure. All calculations included a population of 150 individuals with 27,000 generations and 250,000 energy evaluations. The validation protocol was performed by docking of meloxicam into the crystal structures of cyclooxygenases and the comparison of its position with a crystal structure. 

Binding modes of designed compounds were visualized using Chimera and LIGPLOT v.4.5.3 programs [71,72]. The simulations of biological properties were performed using the 3D/4DQSAR BiS/MC and CoCon algorithms from ChemoSophia Company [73,74].

## 4. Conclusions

The newly designed N-substituted 3,4-pyrroledicarboximides derivatives were tested in terms of their inhibitory activity against COX-1 and COX-2 enzymes. Additionally, their cytotoxicity was determined. The analyzed compounds exhibit an inhibitory effect on both enzymes. Taking into account the results of the colorimetric study, we can assume that all compounds inhibit COX-2 more strongly than meloxicam. The results also indicate that **2b**, **2c** derivatives proved to be the most effective towards the COX-2 enzyme. The **2h** compound, which is the weakest of all studied structures, inhibits the COX-1 enzyme, with COX-2 inhibition value higher than the reference drug, also deserves attention.

The molecular docking studies indicate that all designed compounds can bind to the hydrophobic pocket created by polar and hydrophobic amino acid residues. The hydrophobic and van der Waals forces are the origin of stabilization of protein–ligand complexes. Therefore, compounds of this type can be very promising cyclooxygenase inhibitors; however, they still require further research.

## Data Availability

Not applicable.

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
