# Peer review of "Design and Synthesis of N-Substituted 3,4-Pyrroledicarboximides as Potential Anti-Inflammatory Agents"

_ijms, 2021, doi:10.3390/ijms22031410_

Round 1

Reviewer 1 Report

"It is unacceptable for Supplementary Materials to be on another server that the reviewer cannot access, even after logging in.

Please put all figures, tables, diagrams in the formats required by IJMS in the manuscript so that they constitute one whole.

There is no general recipe for the preparation of compounds (part 2.1). The authors cite syntheses from the previous work (ref. Redzicka et al. [38] from 2019), which gives the impression that the reviewed article is only the completion of the previous one, which means that it lacks a complete novelty.

How many replications were biological tests performed? No deviation would suggest that it is ONLY in one. If indeed in one repetition, please explain how the research was not adjusted to generally accepted rules (at least 3 repetitions)

"All chemicals used were purchased from commercial suppliers." - Please specify which companies and what purity of compounds.

Line 250: silica-gel-60-F254-coated - please write correctly.

Whole Part 3. - please add the producers, countries and cities of the measuring equipment and compounds used.

Part 3.1.2. - merges into one whole. Please sort and divide it somehow. In the ESI-MS analysis [L+H]+ should be written as [M+H]+. Likewise, further ESI-MS measurements. FT-IR measurements were made at KBr? It was not specified.

Line 277: p should in name of compound should be written italics

Line 451: CO2 - please write correctly as CO2.

Article needs to be rebuilt totally in terms of presentation."

Author Response

Manuscript ID: ijms-1067724                                                           Aleksandra Redzicka
Department of Chemistry of Drugs

Faculty of Pharmacy

Wrocław Medical University

Borowska 211

50-556 Wrocław

Poland

24th January 2021

Dear Professor,

We are grateful for all critical comments on the manuscript "Design and synthesis of N-substituted 3,4-pyrroledicarboximides as potential anti-inflammatory agents" We revised our manuscript and resubmitted the new version to the International Journal of Molecular Sciences. All text changes are marked. Moreover, please find our answers below.

We hope that our revised manuscript meets your expectations and thanks to the proposed improvements is now eligible for publication in the International Journal of Molecular Sciences.

Response to Review

According to valuable suggestions of the Referee we have made the following changes to the manuscript:

It is unacceptable for Supplementary Materials to be on another server that the reviewer cannot access, even after logging in.

Supplemental materials have been uploaded to the zenodo website as suggested by the editors. Due to the resulting problems, they will still be attached to the manuscript. We are very sorry for this inconvenience.

Please put all figures, tables, diagrams in the formats required by IJMS in the manuscript so that they constitute one whole.

Figures and tables were included in the manuscript. The supplementary data were added.

There is no general recipe for the preparation of compounds (part 2.1). The authors cite syntheses from the previous work (ref. Redzicka et al. [38] from 2019), which gives the impression that the reviewed article is only the completion of the previous one, which means that it lacks a complete novelty.

In the present paper, the new series of pyrrolo[3,4-c]pyrrole derivatives was designed and synthesized as potential anti-inflammatory agents. The structural modifications involving the introduction of new pharmacophore groups based on N-substituted piperazine moiety to establish the structure–activity relationship. It has been shown in previous studies that the high electron-donating ability of the piperazine ring can be explicitly correlated to the high anti-inflammatory activity of compounds[1]. New compounds were obtained based on the procedure described in our previous study [2]. However, following the reviewer's suggestion not to mislead the reader, this sentence was removed from the manuscript.

  1. D. Hatnapure, A.P. Keche, A.H. Rodge, S.S. Birajdar, R.H. Tale, V.M. Kamble, Synthesis and biological evaluation of novel piperazine derivatives of flavone as a potent anti-inflammatory and antimicrobial agent, Bioorg Med Chem Lett. 2012, 22, 6385–6390.
  2. Redzicka, Ł. Szczukowski, A. Kochel, B. Wiatrak, K. Gębczak, Ż. Czyżnikowska, COX-1/COX-2 Inhibition Activities and Molecular Docking Study of Newly Designed and Synthesized Pyrrolo[3,4-c]Pyrrole Mannich Bases. Bioorganic Med Chem 2019, 27 (17), 3918–3928.

How many replications were biological tests performed? No deviation would suggest that it is ONLY in one. If indeed in one repetition, please explain how the research was not adjusted to generally accepted rules (at least 3 repetitions)

All biological studies were performed in three independent replications. We supplemented the standard deviation in the table and the information on the number of repetitions in materials and methods.

"All chemicals used were purchased from commercial suppliers." - Please specify which companies and what purity of compounds.

We have added relevant information.

Line 250: silica-gel-60-F254-coated - please write correctly.

As suggested, we have made the appropriate corrections.

Whole Part 3. - please add the producers, countries and cities of the measuring equipment and compounds used.

As suggested by the Reviewer, the above information has been included in the manuscript.

Part 3.1.2. - merges into one whole. Please sort and divide it somehow. In the ESI-MS analysis [L+H]+ should be written as [M+H]+. Likewise, further ESI-MS measurements. FT-IR measurements were made at KBr? It was not specified.

[L+H]+ should be written as [M+H]+.  We corrected it.

FT-IR measurements samples  were applied as solids, and frequencies are reported in cm-1

Line 277: p should in name of compound should be written italics

We have corrected it.

Line 451: CO2 - please write correctly as CO2.

We have corrected it.

Article needs to be rebuilt totally in terms of presentation."

As suggested by the Reviewer, fragments of manuscript were reorganized.

Thank you for your kindness in taking the time to consider our work, we await your response with interest.

In the meantime, we remain

Yours faithfully

Aleksandra Redzicka

For, and a behalf of:

Żaneta Czyżnikowska, Benita Wiatrak, Katarzyna Gębczak and Andrzej Kochel

Reviewer 2 Report

The work of A. Redzicka et al “Design and synthesis of N-substituted 3,4-pyrroledicarboximides as a potential anti-inflammatory agent” devoted to the interesting and actual theme. In the present work, authors provide detailed information about the design and synthesis of N-substituted 3,4-pyrroledicarboximides. Despite a fairly good impression about the submitted manuscript I have several minor questions\suggestions to authors:

  1. Throughout the text, abbreviations are used, which should be fully written out on first use or in section Abbreviations.
  2. No reference to the paper of W. Malinka et al. “Synthesis of some N-substituted 3,4-pyrroledicarboximides as potential CNS depressive agents” Pharmazie, 2000 Jan;55(1):9-16.
  3. Missing figure 2 caption.
  4. It seems more logical to present the chapter methods and materials first and then Results and discussion.

From my point of view, the manuscript is suitable for IJMS section: Molecular
Pharmacology, and could be published after minor revision.

Author Response

Manuscript ID: ijms-1067724                                                           Aleksandra Redzicka
Department of Chemistry of Drugs

Faculty of Pharmacy

Wrocław Medical University

Borowska 211

50-556 Wrocław

Poland

24th January 2021

Dear Professor,

We are grateful for all critical comments on the manuscript "Design and synthesis of N-substituted 3,4-pyrroledicarboximides as potential anti-inflammatory agents" We revised our manuscript and resubmitted the new version to the International Journal of Molecular Sciences. All text changes are marked. Moreover, please find our answers below.

We hope that our revised manuscript meets your expectations and thanks to the proposed improvements is now eligible for publication in the International Journal of Molecular Sciences.

Response to Review

According to the valuable suggestions of the Referee we have made the following changes to the manuscript:

Throughout the text, abbreviations are used, which should be fully written out on first use or in section Abbreviations.

We corrected it.

No reference to the paper of W. Malinka et al. “Synthesis of some N-substituted 3,4-pyrroledicarboximides as potential CNS depressive agents” Pharmazie, 2000 Jan;55(1):9-16.

The paper of W. Malinka et al. “Synthesis of some N-substituted 3,4 pyrroledicarboximides as potential CNS depressive agents” Pharmazie, 2000 Jan;55(1):9-16 was introduced as a reference [32]

Missing figure 2 caption.

The caption has been added.

It seems more logical to present the chapter methods and materials first and then Results and discussion.

The order of chapters is suggested by the publisher. However, to improve the readability of the manuscript, we have made some changes, which are indicated in the attached file.

Thank you for your kindness in taking the time to consider our work, we await your response with interest.

In the meantime, we remain

Yours faithfully

Aleksandra Redzicka

For, and a behalf of:

Żaneta Czyżnikowska, Benita Wiatrak, Katarzyna Gębczak and Andrzej Kochel

Reviewer 3 Report

A list of the compounds synthesized is simply missing. It makes it difficult to relate to any of the other information given!

On page 6 in section 2.4, the authors use the L & V-rules to rule out the compound 2b as a drug candidate in spite of the compound having the best selectivity between COX-1 and COX-2. May I remind about drugs like Atorvastatin, Naproxen or Indomethacin?

The reference list do also need a little correction with respect to the format, where some titles are in capital letters.

Author Response

Manuscript ID: ijms-1067724                                                           Aleksandra Redzicka
Department of Chemistry of Drugs

Faculty of Pharmacy

Wrocław Medical University

Borowska 211

50-556 Wrocław

Poland

24th January 2021

Dear Professor,

We are grateful for all critical comments on the manuscript "Design and synthesis of N-substituted 3,4-pyrroledicarboximides as potential anti-inflammatory agents" We revised our manuscript and resubmitted the new version to the International Journal of Molecular Sciences. All text changes are marked. Moreover, please find our answers below.

We hope that our revised manuscript meets your expectations and thanks to the proposed improvements is now eligible for publication in the International Journal of Molecular Sciences.

Response to Review

According to the valuable suggestions of the Referee we have made the following changes to the manuscript:

A list of the compounds synthesized is simply missing. It makes it difficult to relate to any of the other information given!

The list of compounds has been added.

On page 6 in section 2.4, the authors use the L & V-rules to rule out the compound 2b as a drug candidate in spite of the compound having the best selectivity between COX-1 and COX-2. May I remind about drugs like Atorvastatin, Naproxen or Indomethacin?

We agree with the Reviewer that our statement regarding the potential application of compound 2b as a drug should be clarified. We added some extension sentences on page 7 and hope that now it is more clear for a potential reader. Please find below the introduced changes in the manuscript.

2.3.2 SAR of N-substituted 3,4-pyrroledicarboximides 2a-2p

The SAR of N-substituted 3,4-pyrroledicarboximides (Table 2) shows the following:

  • all compounds 2a-2p inhibited COX-2 a stronger than meloxicam, which was used as a reference
  • within the piperazine derivatives, the best COX selectivity ratio IC50 (COX-2)/IC50 (COX-1) had compounds 2b and 2c. Unfortunately, Compound 2b does not meet the Lipinski and Veber rules. However, it should be emphasized, that these rules define only general properties that describe potential orally administered drugs. Therefore, the predictive ability of these rules may be limited in some cases and one should be careful during the interpretation of such types of results.

The reference list do also need a little correction with respect to the format, where some titles are in capital letters.

The revision of the reference list has been done.

Thank you for your kindness in taking the time to consider our work, we await your response with interest.

In the meantime, we remain

Yours faithfully

Aleksandra Redzicka

For, and a behalf of:

Żaneta Czyżnikowska, Benita Wiatrak, Katarzyna Gębczak and Andrzej Kochel

Round 2

Reviewer 1 Report

Thank you for all the explanations and the corrected manuscript sent.

I would suggest in Table 2, for example, "87.90 ± 0.03", instead of "87.90 (0.03)". As the authors themselves write in paragraph 3.3.5 "IC50 values ± SD". Additionally, I would suggest re-editing paragraph 3.1.2 to make it more readable as it is still hard to read - no clearly separated parts.
I hope that finally the Supplementary Materials part will be available as one or several files, not just so many files, what make finding

Following these suggestions at the final editing stage, the article could be published in IJMS MDPI.

Author Response

Manuscript ID: ijms-1067724                                                                      Aleksandra Redzicka
Department of Chemistry of Drugs

Faculty of Pharmacy

Wrocław Medical University

Borowska 211

50-556 Wrocław

Poland

27th January 2021

Dear Professor,

We revised our manuscript and resubmitted the new version to the International Journal of Molecular Sciences. All text changes are marked. Responses to comments are provided below.:

I would suggest in Table 2, for example, "87.90 ± 0.03", instead of "87.90 (0.03)". As the authors themselves write in paragraph 3.3.5 "IC50 values ± SD". Additionally, I would suggest re-editing paragraph 3.1.2 to make it more readable as it is still hard to read - no clearly separated parts.

As suggested by the Reviewer, Table 2 and paragraph 3.1.2 have been corrected

I hope that finally the Supplementary Materials part will be available as one or several files, not just so many files, what make finding

The relevant files in the Supplementary Materials were merged, all compressed, and reattached to the manuscript.

Yours faithfully

Aleksandra Redzicka

For, and a behalf of:

Żaneta Czyżnikowska, Benita Wiatrak, Katarzyna Gębczak and Andrzej Kochel